# Genome-Wide Association Study of Diabetogenic Adipose Morphology in the GENetics of Adipocyte Lipolysis (GENiAL) Cohort

**DOI:** 10.3390/cells9051085

**Published:** 2020-04-27

**Authors:** Veroniqa Lundbäck, Agné Kulyté, Peter Arner, Rona J. Strawbridge, Ingrid Dahlman

**Affiliations:** 1Lipid laboratory, Endocrinology Unit, Department of Medicine Huddinge, Karolinska Institutet, 171-77 Stockholm, Sweden; veroniqa.lundback@ki.se (V.L.); agne.kulyte@ki.se (A.K.); peter.arner@ki.se (P.A.); 2Institute of Health and Wellbeing, University of Glasgow, College of Medicine, Veterinarian and Life Sciences, Glasgow G12-8RZ, UK; Rona.Strawbridge@glasgow.ac.uk; 3Department of Medicine Solna, Karolinska Institutet, 171-77 Stockholm, Sweden; 4Health Data Research University of Glasgow, College of Medicine, Veterinarian and Life Sciences, Glasgow G12-8RZ, UK

**Keywords:** adipose morphology, adipogenesis, genome-wide association study (GWAS), genetic loci, GENetics of Adipocyte Lipolysis (GENiAL) cohort

## Abstract

An increased adipocyte size relative to the size of fat depots, also denoted hypertrophic adipose morphology, is a strong risk factor for the future development of insulin resistance and type 2 diabetes. The regulation of adipose morphology is poorly understood. We set out to identify genetic loci associated with adipose morphology and functionally evaluate candidate genes for impact on adipocyte development. We performed a genome-wide association study (GWAS) in the unique GENetics of Adipocyte Lipolysis (GENiAL) cohort comprising 948 participants who have undergone abdominal subcutaneous adipose biopsy with a determination of average adipose volume and morphology. The GWAS identified 31 genetic loci displaying suggestive association with adipose morphology. Functional evaluation of candidate genes by small interfering RNAs (siRNA)-mediated knockdown in adipose-derived precursor cells identified six genes controlling adipocyte renewal and differentiation, and thus of potential importance for adipose hypertrophy. In conclusion, genetic and functional studies implicate a regulatory role for *ATL2*, *ARHGEF10*, *CYP1B1*, *TMEM200A, C17orf51,* and *L3MBTL3* in adipose morphology by their impact on adipogenesis.

## 1. Introduction

Excess adipose tissue plays a central role in the risk of developing insulin resistance (IR) and, thereby, the development of type 2 diabetes (T2D). However, it is not only the size but also the morphology of subcutaneous adipose tissue (SAT), which influences adipose function [1,2]. The number of fat cells differs substantially between individuals, even among those with the same total body fat, giving rise to hypertrophic, i.e., fewer but larger fat cells, or hyperplastic, i.e., more but smaller fat cells, adipose morphology [3,4]. Hypertrophic SAT is the pernicious morphology and has been associated with IR and T2D [5,6]. Hypertrophic SAT has been linked to local IR and increased release of pro-inflammatory mediators and free fatty acids within SAT [5,6], which through systemic effects, contribute to systemic IR.

The factors governing SAT hyperplastic versus hypertrophic expansion is poorly understood. Human SAT contains a pool of progenitor cells that are continuously recruited to the adipose lineage to mature adipocytes [3]. Hypertrophic adipose morphology has been associated with reduced numbers of new adipocytes [4,6] and altered extracellular matrix function [7]. Adipocytes are turned over slowly [4], and adipose morphology is, thus, a relatively stable phenotype over time. Importantly, the observation that bariatric surgery induces a hyperplastic adipose morphology, which is associated with beneficial systemic metabolic parameters in comparison to body mass index (BMI)-matched controls, gives support to the notion that treatments improving adipose morphology have potential as therapeutic targets for IR and T2D [8].

First degree relatives of patients with T2D have been characterized as having hypertrophic adipose tissue consistent with a genetic impact on adipose morphology and link to IR/T2D [9]. Recently, large genome-wide association studies (GWAS) have identified hundreds of genetic risk loci for IR and T2D, and have reported that subjects carrying certain risk alleles exhibit reduced SAT indicating that inability of proper SAT expansion and body fat distribution are important contributing factors for the risk of developing metabolic disease [10]. There is evidence that body fat distribution and adipose morphology are related [7]. However, to the best of our knowledge, up to now, no GWAS has directly addressed the genetic regulation of adipose morphology, and its role in development of IR/T2D.

In the present study, we conducted a GWAS for adipose morphology to identify genetic loci associated with, and highlight potential variants influencing adipose morphology. Adipose morphology was assessed using average adipocyte size in the abdominal SAT region adjusted for the size of the fat depot; a positive value is indicative of larger than expected adipocytes in relation to body fat mass (hypertrophy), and a negative value indicates hyperplasia [11]. Candidate genes in adipose morphology-associated loci were taken forward for functional evaluation, with in vitro experiments in adipocytes being used to determine the impact on phenotypes central to the development of adipose hypertrophy, i.e., precursors cell proliferation and adipocyte differentiation. 

## 2. Materials and Methods

### 2.1. Participants

The GENiAL cohort has been described previously [12]. In the present study, we included all 948 participants for which SAT adipocyte size had been measured to determine adipose morphology (Appendix A). Descriptives of the study subjects are presented in Table 1. Fifty-seven percent were obese (defined as BMI ≥ 30kg/m^2^). Total body fat mass was indirectly calculated using a formula based on age, sex, and BMI [13]. A venous blood sample was obtained for the extraction of DNA and clinical chemistry analyses by the hospital’s accredited routine clinical chemistry laboratory. All participants lived in Stockholm county, Sweden. One hundred and ninety-four participants had type 2 diabetes, hypertension, or dyslipidemia alone or in different combinations. None were treated with insulin, glitazones, or glucagon-like-peptide analogs. Data on clinical and adipose variables have been published previously [14,15]. The study was approved by the local committee on ethics at the Huddinge University Hospital (D. no. 167/02, 2002-06-03) and explained in detail to each participant. Informed consent was obtained from all participants; this was in written form since 1996.

SAT gene expression was studied in 114 women, which is a subset of the above cohort. This cohort has been described previously [16] and contained the same type of adipose data as presented herein, with the exception that adipose morphology was calculated by relating average adipose volume to total body fat.

### 2.2. Adipose Tissue Biopsy and Measurement of Adipose Morphology

GENiAL participants were examined in the morning after an overnight fast. Following a clinical examination, an abdominal SAT biopsy was obtained by a needle aspiration biopsy lateral to the umbilicus as described [17]. SAT samples were rapidly rinsed in sodium chloride (9 mg/mL) before the removal of visual blood vessels and cell debris and subsequently subjected to collagenase treatment to obtain isolated adipocytes as described [18]. The mean weight and volume of adipocytes and their spontaneous lipolysis were determined as described [19,20]. A curve fit of the relationship between mean adipocyte volume and estimated abdominal subcutaneous fat mass (ESAT) was performed exactly as described [11]. This non-linear relationship for the GENiAL cohort is published in Andersson D et al. The difference between the measured and expected mean adipocyte volume at the corresponding fat mass determines adipose morphology [11]. If the measured adipocyte volume is larger than expected, SAT hypertrophy prevails, whereas the opposite is valid for hyperplasia. Thus, this measure of adipose morphology is independent of fat mass.

### 2.3. Genetic Analysis

Genotyping and quality control (QC) have been described [12]. After quality control, imputation was performed using the haplotype reference consortium panel and, when variants were not available, using the 1000G phase3 reference panel [21]. Post-imputation quality control excluded single nucleotide polymorphisms (SNPs) with minor allele count <3 and imputation quality information INFO <0.4 as well as related individuals (one of each pair of 1^st^ or 2^nd^-degree relatives). After quality control, 894 samples and 9,714,326 SNPs were available for phenotypic analysis. A GWAS was conducted in PLINK 2.0 [22] using linear regression, assuming an additive genetic model, and adjusting for population structure (PCs1-3), age, and sex. Genome-wide significance was set at *p* < 5 × 10^−8,^ and suggestive significance was set at *p* < 1 × 10^−5^. Only SNPs with minor allele frequency (MAF) >1% were included in the results. Results were visualized using FUMA [23]. Data for one SNP included in the herein described GWAS is included in a separate manuscript by Carl Herdenberg E et al., where leucine-rich repeats and immunoglobulin-like domains 1 (LRIG) proteins regulate lipid metabolism via bone morphogenic (BMP) signaling and affect the risk of type 2 diabetes (submitted). This SNP displays a nominal association only with adipose morphology and is, therefore, not included in the presentation of GWAS results in the present study.

### 2.4. Data Mining

SNPs reaching GWAS or suggestive significance were examined for genotype-specific gene expression in adipose tissue (expression quantitative locus; eQTL) using GTEx database on January 15, 2020 [24]. 

### 2.5. Adipocyte Cell Culture and Transfection With Small Interfering RNA

Isolation, growth, and differentiation of SAT-derived human mesenchymal stem cells (hMSCs) have been described previously [25]. Marker gene expression (Cap analysis gene expression (CAGE) and gene expression data) and lipid accumulation images for these cells have been published previously [26]. 

hMSCs were transfected using a Neon electroporator (Invitrogen, Carlsbad, CA, US) according to the manufacturer’s protocol. Briefly, 1 million hMSCs at day −4 before initiation of differentiation were mixed with 40 nM ON-TARGETplus SMARTpool small interfering RNAs (siRNAs) targeting genes of interest or non-targeting siRNA pool (Dharmacon, Lafayette, CO, US) and electroporated using a 100 µL NEON electroporation tip. Electroporation conditions were 1150 Volts, 30 ms width, 2 pulses. Electroporation was repeated until the required number of cells were collected for the experimental set-up. Following electroporation, the cells were plated in an antibiotic-free medium at the density of 55,000 cells/well in 24-well plates or 5500 cells/well in 96-well plate and cultured up to day 9 of differentiation as specified below. 

### 2.6. Measurement of Lipid Accumulation and Lipolysis

Lipid accumulation was quantified at differentiation day 9. hMSCs differentiated in vitro were washed with PBS and fixed with 4% paraformaldehyde solution for 10 min at room temperature. Following fixation, neutral lipids were stained with Bodipy 493/503 (at 0.2 µg/mL; Molecular Probes, Thermo Fisher Scientific, Waltham, MA, US), and nuclei (DNA) were stained with Hoechst 33342 (at 2 µg/mL; Molecular Probes) for 20 min at room temperature. Accumulation of neutral lipids and cell numbers were quantified in CellInsight™ CX5 High Content Screening (HCS) Platform (Thermo Fischer Scientific, Waltham, MA, US) with integrated “Spot detection” protocol. Total Bodipy fluorescence (lipid droplets) was normalized to the number of nuclei, representing the number of cells in each well. 

The medium was collected at day 7 and 9 of differentiation of hMCSs for measuring glycerol release as an index of lipolysis, as described [27]. A standard curve ranging from 0 to 120 µM was used to calculate the concentrations of the samples. Amounts of glycerol was normalized to the number of nuclei and in each well.

### 2.7. Analysis of Cell Proliferation 

hMSCs were electroporated using 10 µl NEON electroporation tip day −4 before initiation of differentiation. Electroporation conditions were 1150 Volts, 30 ms width, 2 pulses. Cells were incubated with media containing 5 µM 5-ethynyl-2′-deoxyuridine (EdU) at day −3 for 24 h, followed by an assessment of EdU-positive cells using the Click-iT^®^Plus EdU Alexa Fluor555 (C10352, Invitrogen, Carlsbad, CA).

### 2.8. Isolation of RNA and Analysis of Gene Expression

Following electroporation of siRNA, hMSCs were collected for isolation of RNA at the day differentiation was induced (day 0), days 1, and 7 post-induction of differentiation. Extraction of total RNA, measurement of concentration and purity as well as reverse transcription were performed as described [27]. Quantitative RT-PCR of coding genes was performed using commercial TaqMan probes (Applied Biosystems, Foster City, US). Gene expression was normalized to the internal reference gene 18s. Relative expression was calculated using the 2(−ΔΔ threshold cycle) method [28].

### 2.9. Statistical Analysis of Clinical and In Vitro Data

Clinical and in vitro results are presented as mean ± standard deviation (SD). Glycerol and triglycerides values were normalized by log10 transformation prior to analysis. Standard statistical tests were used, including *t*-test, single or multiple regression as indicated in figure/table legends using Stat View software (Abacus Concepts Inc, Berkley, CA) or JMP 14 (SAS, NC, USA). 

## 3. Results

### 3.1. Clinical Findings

The demographic characteristics of the GENiAL cohort participants included in the GWAS of adipose morphology are presented in Table 1. The cohort consisted predominantly of women, who were younger and had a higher frequency of obesity than the men. Obesity was associated with higher systemic levels of fasting glucose and insulin, and with a pernicious lipid profile in both sexes. Average abdominal subcutaneous adipocyte volume was larger in obese subjects, whereas adipose morphology, as expected, was unrelated to obesity status.

Adipose morphology was correlated with insulin resistance as estimated by fasting serum (fS)-Insulin (standardized beta 0.24, *p* 2 × 10^−12^), and HOMA-IR (standardized beta 0.16, *p* 1.5 × 10^−6^) after adjusting for age and sex.

### 3.2. GWAS for Adipose Morphology

The results of the GWAS are presented in Appendix A, Table 2, and Appendix A. A total of 66 SNPs in 31 loci showed suggestive association (*p* < 10^−5^) with adipose morphology. Six genetic loci, each <1 Mb, contained ≥ 3 SNPs displaying suggestive association (*p* < 10^−5^) with adipose morphology (Figure 1); subsequent analysis was focused on these genomic regions. The strongest association with adipose morphology was observed in a region on chromosome 3, where one SNP, rs2378515, displayed a borderline GWAS-significant association (*p* 5.20 × 10^−8^). This SNP is situated in the *TP63* gene (Figure 1), which was not expressed in mature adipocytes, nor in our hMSCs during in vitro differentiation (results not shown) and could, therefore, not be taken forward for functional evaluation. No other gene is encoded within this LD-block bounded by a recombination fraction ~20% (Figure 1). On chromosome 2, we identified tag-SNP rs147711728 among multiple SNPs (*n* = 6) of suggestive significance in the region that harbors the *ATL2*, *CYP1B1,* and *RMD2* genes. On chromosome 6, tag-SNP rs62431222 defines a locus, which includes eight SNPs reaching suggestive significance and harboring genes *L3MBTL3, SAMD3,* and *TMEM200A*. Chromosome 8, with tag-SNP rs11988258 and three other SNP with suggestive association, had genes *CLN8* and *ARHGEF10* in the locus. On chromosome 9, tag-SNP rs145072648 is in a locus containing the *KLHL9* and the *IFNA* family of genes. The chromosome 17 locus includes three associated SNPs with tag-SNP rs201766885 and harbors *FAM27L, MTRNR2L1,* and *C17orf51*. 

All 66 SNPs in Appendix A were investigated in the GTEx eQTL resource, but failed to provide evidence for genotype-dependent gene expression in SAT. All SNPs resided in regions that do not encode protein and are, therefore, of unclear impact.

### 3.3. Functional Evaluation of Candidate Genes

Transcriptome profiling of all protein-coding genes in the six adipose morphology-associated genetic loci revealed six genes from four loci, which were expressed in adipocytes and their progenitors isolated from SAT biopsies, as well as in hMSCs undergoing in vitro differentiation to adipocytes. (Figure 2A,B). The six genes were taken forward for functional evaluation in hMSCs to determine possible impact on adipogenesis. All genes except *C17orf51* were enriched in progenitor cells as compared to mature adipocytes (Figure 2A). The expression of *CYP1B1*, *TMEM200A,* and *ARHGEF10* decreased directly after induction of differentiation, whereas the expression of *ATL2*, *C17orf51,* and *L3MBTL3* varied over the course of differentiation of hMSCs (Figure 2B). 

Based on the expression pattern during differentiation, we knocked down the aforementioned candidate genes in hMSCs during proliferation, that is four days prior to induction of differentiation where day 0 corresponds to the start of differentiation. This resulted in a 60–80% decreased expression in all candidate genes except *L3MBTL3,* where the efficiency of knockdown was 20–60% (Figure 3A). We evaluated the accumulation of neutral lipids as a marker of adipocyte differentiation (Figure 3B, Appendix A) and the release of glycerol in cell culture medium as a marker of adipocyte lipolysis at day 9 of differentiation (Figure 3C). Both the accumulation of lipids and glycerol levels were reduced after knockdown of all six genes. We observed a ~5–10% increase in cell number in knockdown of *ATL2*, *ARHGEF10, CYP1B1,* and *TMEM200A* and a modest reduction in cell number in knockdown of *L3MBTL3* (Figure 4A). To investigate cell proliferation, we knocked down the candidate genes, and treated cells in each experiment with EdU, a marker of DNA synthesis in proliferating cells. The proportion of proliferating cells (EdU^+^) was ~5–16% higher for all genes, although for *L3MBTL3,* the increase was modest (Figure 4B); however, this might be related to poor knockdown efficiency of *L3MBTL3* (Figure 3A). Furthermore, the expression of the marker of proliferation Ki-67 (*mKi67)* was increased day 0 before initiation of differentiation for all investigated genes (Figure 4C). 

Expression levels of genes central to adipogenesis (peroxisome proliferator-activated receptor-gamma (*PPARG*), CCAAT/enhancer-binding protein alpha (*CEBPA*) and CCAAT/enhancer-binding protein beta (*CEBPB*) and the adipocyte marker adiponectin (*ADIPOQ*) were measured in each knockdown experiment (Figure 5) at differentiation start (day 0), day 1 day, and day 7 post differentiation induction. Knockdown of *ATL2* resulted in a reduced expression of all adipogenesis genes (Figure 5A). *ARHGEF10* knockdown resulted in a temporary modest reduction in the expression of *CEBPA* and *CEBPB* (Figure 5B). The knockdown of *CYP1B1* resulted in reduced expression of *PPARG*, *CEBPB,* and *ADIPOQ,* and a modest reduction in *CEBPA* expression (Figure 5C). The knockdown of *TMEM200A* and *L3MBTL3* resulted in a reduced expression of investigated adipogenesis genes, although the effect on *PPARG* expression was modest (Figure 5D,F). The knockdown of *C17orf51* resulted in a strong reduction in *ADIPOQ* expression and a modest effect on the expression of adipogenesis genes (Figure 5E).

### 3.4. SAT-Expression of CYP1B1 and ATL2 Correlates With Adipose Morphology

We next examined whether abdominal SAT expression of the six candidate genes was associated with adipose morphology using SAT-expression from a previously examined cohort of 114 women where adipose morphology was quantified as average adipocyte size in the abdominal SAT region adjusted for total body fat [16]. There was an inverse relationship between *ATL2* expression and adipose morphology (*p* 4.00 × 10^−4^, R = 0.33) and a positive association between *CYP1B1* and adipose morphology (*p* 3.01 × 10^−6^, R = 0.43). No association between *ARHGEF10*, *TMEM200A*, *L3MBTL3,* and *C17orf51* and adipose morphology was observed.

### 3.5. Overlap Between Result From GWAS Studies of T2D and Adipose Morphology

None of the SNPs demonstrating suggestive association with adipose morphology have previously been associated with clinical metabolic traits, such as IR or T2D, according to the GWAS catalog (https://www.ebi.ac.uk/gwas/), February 14th, 2020). However, 25 genetic loci previously associated with T2D demonstrated nominal association with adipose morphology (Table 3). In 12 genetic loci, the allele increasing the risk of T2D was associated with adipose hypertrophy consistent with the finding that this adipose morphology increases the risk for T2D in genetic epidemiological studies [9]. None of these genetic loci encoded genes with established roles in human SAT. For the remaining loci, the risk allele is either not reported in the GWAS catalog, or the allele increasing the risk of T2D is associated with hyperplastic SAT making it difficult to interpret the finding. 

## 4. Discussion

The present study has shed new light on the genetics of adipose morphology, identified six genetic loci harboring candidate genes, and provides evidence for these candidate genes as regulators of adipocyte renewal and lipid storage in adipose hypertrophy. Hypertrophic SAT morphology is a strong risk factor for IR and T2D. However, among established risk alleles for T2D from GWAS, only 12 alleles were nominally associated with hypertrophic adipose morphology.

In the adipose morphology-associated locus on chromosome 2, functional analysis by siRNA-mediated knockdown suggested *CYP1B1* and/or *ATL2* as potential causal links between genetic variants and adipose morphology. The knockdown of either gene stimulated proliferation of adipose-derived hMSCs accompanied by reduced expression of mRNA markers for adipogenesis, inhibited lipid accumulation, and reduced lipolysis in cells. Our interpretation of these findings is that a higher number of precursor cells resulted in a lower proportion of differentiating cells, i.e., reduced expression of adipogenesis markers and reduced lipid accumulation, rather than impaired adipogenesis. This interpretation is consistent with the finding that increased recruitment of new fat cells is a major determinant of adipose hyperplasia [4]. The expression pattern of *CYP1B1* and *ATL2* did not permit knockdown late in differentiation and, therefore, we could not determine whether these genes have a direct impact on mature fat cell lipid accumulation, or whether the lipid phenotype is secondary to a larger number of progenitor cells in the experiment. In either case, our findings support the hypothesis that *CYP1B1* and *ATL2* counteract a favorable hyperplastic adipose expansion by inhibiting the proliferation of precursor cells. The evidence for *CYP1B1* was further supported by the finding of a positive association between SAT gene expression of *CYP1B1* and hypertrophic adipose morphology in a large clinical cohort*. CYP1B1* encodes a widely expressed monooxygenase involved in the metabolism of many important physiological compounds, including estrogen, arachidonic acid, melatonin, and retinoids [30]. *CYP1B1*-null mice display attenuated high-fat diet (HFD)-induced obesity and improved glucose tolerance related to increased fatty acid oxidation markers [31,32,33]. In mouse models, *CYP1BI* has been reported to control *PPARG* expression, whereas no impact on adipogenesis was observed [30]. The other candidate gene on chromosome 2, *ATL2,* has been reported to influence lipid droplets via the endoplasmatic reticulum [34]. The chromosome 2 locus, but not the SNPs reported herein, has previously been associated with BMI in GWAS [35]. Together previous and present findings give support to the notion that the chromosome 2 locus, possible via *CYP1B1* and/or *ATL2*, controls the extent and fate of adipose expansion.

On chromosome 3, multiple SNPs in an intron of the *TP63* gene displayed association with adipose morphology. Due to the high recombination in this locus, the association does not extend beyond the *TP63* gene. *TP63*^-/-^ mice develop obesity and insulin resistance linked to increased fatty acid synthesis and decreased fatty acid oxidation [36]. None of the reported *TP63* splice variants were detected at mRNA level in our in vitro system (adipose derived hMSCs) nor in mature adipocytes isolated from adipose tissue, and, therefore, we could not assess the function of *TP63* in human adipocytes. Further characterization of the potential role of TP63 in human adipose tissue will need more refined methods that were beyond the scope of this study. 

In the chromosome 6 locus, knockdown of *L3MBTL3* and *TMEM200A* reduced lipid accumulation in fat cells. For *TMEM200A,* the finding was accompanied by the proliferation of adipose-derived hMSCs. The *L3MBTL3* locus has recently been reported to be associated with systemic IR and lower levels of peripheral fat, and the encoded protein to stimulate adipocyte fat storage [10]. The herein reported association of the *L3MBTL3* with hypertrophic adipose morphology potentially expands the knowledge of how this locus influences IR. We did not find any previous data on *TMEM200A* and metabolic disease; additional studies are needed to determine the importance of this gene for adipose function.

Finally, loci on chromosomes 8 and 17 were associated with adipose morphology. In these loci, knockdown of the *ARHGEF10* and *C17orf51* genes, respectively, promoted precursor cell proliferation and inhibited lipid accumulation, which could explain the association between these genetic loci and adipose morphology. *ARHGEF10* belongs to the Rho guanine nucleotide exchange factor (GEF) family, which stimulates Rho GTPases and has been shown to have tumor suppressor activity and inhibits proliferation [37]. Beyond this, the function of these genes is poorly defined, and these genetic loci have not previously been implicated in the control of common metabolic disorders according to the GWAS catalog (https://www.ebi.ac.uk/gwas/).

SAT adipose morphology, as measured herein, categorizes the studied adipose depot as hypertrophic or hyperplastic, i.e., larger or smaller fat cells than expected given the size of the fat depot. In general, both average adipocyte volume and numbers increase with increasing BMI. An advantage with the applied measure of adipose morphology is that it takes the size of the fat depot into account. At the same time, adipose morphology, hereby, becomes a complex phenotypic outcome potentially dependent on genes controlling adipocyte number, lipid droplet size, as well as total body fat. In the present study, we chose to focus on precursor cell proliferation and lipid droplet size as potential functional links between candidate genes and adipose morphology. An appropriate functioning adipogenesis is essential for healthy lipid storage in subcutaneous AT. Inappropriate AT expansion, as observed with hypertrophic SAT, promotes ectopic lipid storage and systemic IR [10]. The reduced glycerol release observed after knockdown of candidate genes gives further support to the notion of impaired adipogenesis. The complexity of the phenotype might explain why most examined genes did not show an association between adipose gene expression and morphology.

One weakness of the present study is that we were unable to link SAT morphology associated SNPs functionally to specific genes, i.e., none of the morphology-associated SNPs were eQTLs in SAT or visceral adipose tissue or caused direct deleterious effects on gene function, e.g., missense mutations, according to bioinformatic analysis. Another weakness is the limited power of the study cohort. The latter is unavoidable for a phenotype, such as adipose volume, which is very resource-demanding to collect. However, the results of the functional exploration of genes in the identified morphology-associated loci provide validation for the approach. Finally, we only assessed SAT morphology although other adipose depots might be even more relevant to study [38], but are even less accessible for sampling.

In conclusion, our genetic and functional findings provide evidence for *ATL2*, *ARHGEF10*, *CYP1B1*, *TMEM200A*, *C17orf51,* and *L3MBTL3* being important for adipose tissue morphology by regulation of proliferation and differentiation of precursor cells. Further work would be required to determine whether modulation of these genes could have therapeutic applications in IR and T2D, where adipogenesis is perturbed

## Figures and Tables

**Figure 1 cells-09-01085-f001:**
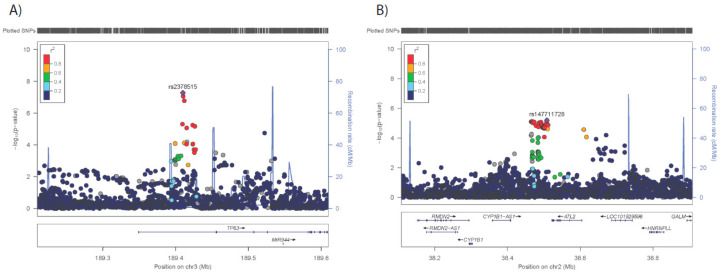
Genetic loci demonstrating suggestive association with adipose morphology.

**Figure 2 cells-09-01085-f002:**
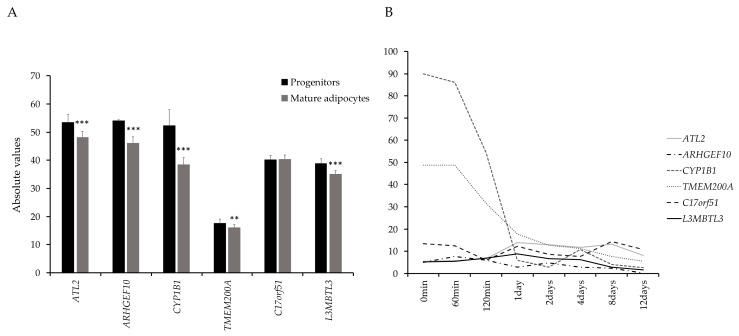
(**A**) Enrichment of candidate genes in progenitor cells and mature adipocytes. Data retrieved from the FANTOM5 dataset (http://fantom.gsc.riken.jp/5/) and Ehrlund, A et al. 2017 [29]. (**B**) Expression of candidate genes during human mesenchymal stem cell (hMSC) differentiation in vitro.

**Figure 3 cells-09-01085-f003:**
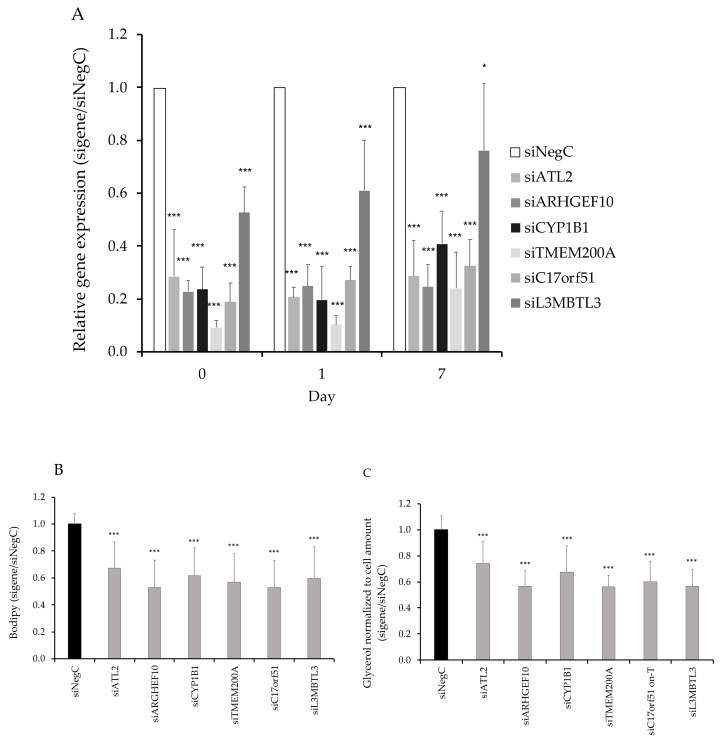
(**A**) *ATL2, ARHGEF10, CYP1B1, TMEM200A, C17orf51, and L3MBTL3* was knocked down using small interfering RNAs (siRNA) in hMSCs four days (day −4) before induction of differentiation and followed until differentiation day 7 and 9. (**B**) Accumulation of neutral lipids was evaluated with Bodipy staining of cells at day 9. (**C**) Glycerol amount in the medium was measured, and all measured concentrations were within the range of the glycerol standard curve 0–120 µM. Results were analyzed using a *t*-test and presented in fold change ± SD relative to negative control of a corresponding time point (Neg C). ****p* < 0.005, ***p* < 0.01, **p* < 0.05.

**Figure 4 cells-09-01085-f004:**
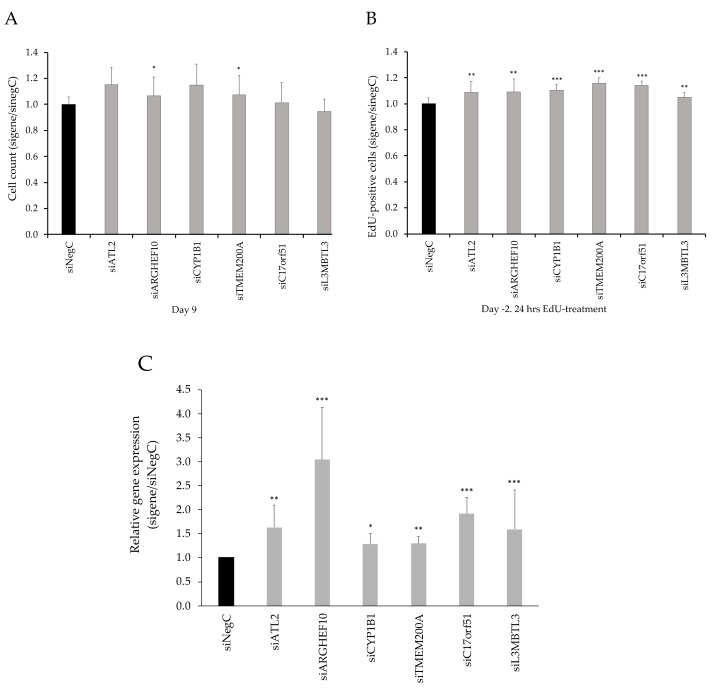
(**A**) Cell number at day 9 of differentiation was measured by Hoechst staining of nuclei. (**B**) Cell proliferation was evaluated by measuring DNA-synthesis with 5-ethynyl-2′-deoxyuridine (EdU). (**C**) mRNA-levels of proliferation marker *mKI67* were measured with real-time qPCR at day 0 before differentiation was initiated. Results were analyzed using a *t*-test and presented in fold change ± SD relative to negative control of a corresponding time point (Neg C). ****p* < 0.005, ***p* < 0.01, **p* < 0.05.

**Figure 5 cells-09-01085-f005:**
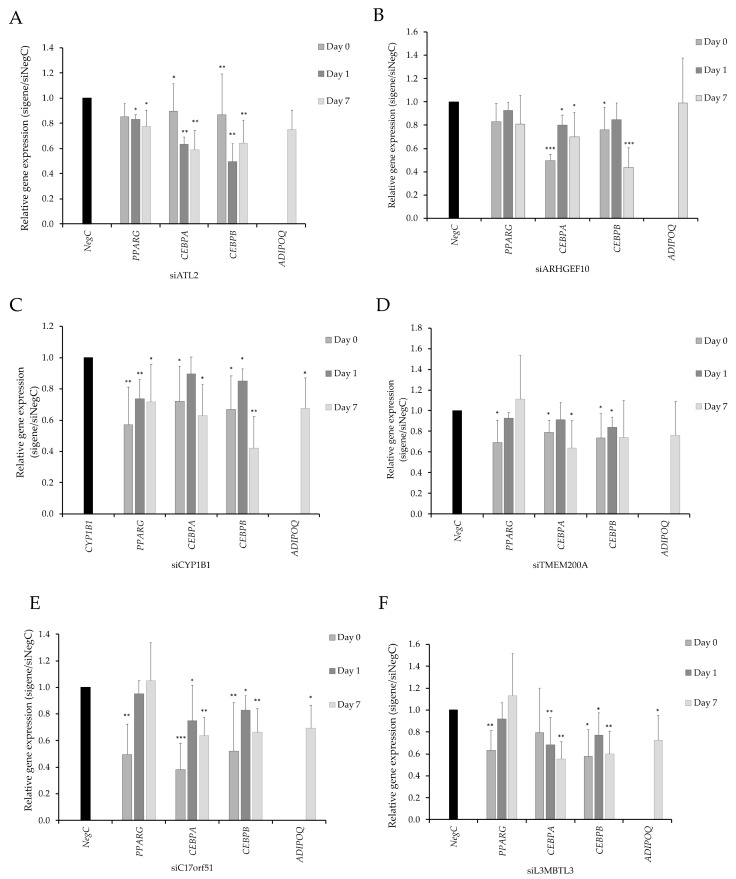
(**A–F**) The expression of peroxisome proliferator-activated receptor-gamma (*PPARG*), CCAAT/enhancer-binding protein alpha (*CEBPA*), and CCAAT/enhancer-binding protein beta (*CEBPB*) measured on day 0, 1, and 7. The expression of *ADIPOQ* was only measured at day 7. Results are based on three biological/independent experiments. Expression of genes was normalized to the reference gene 18s. Results were analyzed using a *t*-test and presented in fold change ± SD relative to negative control of a corresponding time point (Neg C). ****p* < 0.005, ***p* < 0.01, **p* < 0.05.

**Table 1 cells-09-01085-t001:** Characteristics of participants in the GENetics of Adipocyte Lipolysis (GENiAL) cohort used for the analysis of adipose morphology.

	Men			Women		
	Non Obese	Obese	*P*	Non Obese	Obese	*P*
lean/obese (n)	155	91		261	441	
age (years)	47(14)	43(12)	0.01	40(13)	41(10)	0.38
height (cm)	179(6)	181(7)	0.014	167(6)	166(7)	0.0044
body weight (kg)	81(9)	125(18)	<0.0001	67(9)	108(17)	<0.0001
BMI (kg/m^2^)	25(2)	38(5)	<0.0001	24(3)	39(5)	<0.0001
waist (cm)	93(8)	126(13)	<0.0001	84(10)	119(13)	<0.0001
WHR	0.95(0.06)	1.05(0.05)	<0.0001	0.86(0.07)	0.96(0.07)	<0.0001
systolic blood pressure (mm Hg)	129(15)	140(18)	<0.0001	120(16)	130(16)	<0.0001
diastolic blood pressure (mm Hg)	79(10)	86(12)	<0.0001	74(10)	79(10)	<0.0001
plasma total cholesterol (mmol/L)	5.2(1.3)	5.3(1.4)	0.4	4.9(1.1)	5.0(1.0)	0.13
plasma HDL cholesterol (mmol/L)	1.3(0.4)	1.0(0.2)	<0.0001	1.6(0.4)	1.2(0.3)	<0.0001
plasma triacylglycerides (mmol/l)	1.72(2.1)	2.36(2.52)	0.044	1.07(0.64)	1.53(0.87)	<0.0001
plasma non-esterified fatty acids (mmol/L)	0.49(0.17)	0.61(0.21)	<0.0001	0.62(0.21)	0.72(0.24)	<0.0001
plasma glycerol (mmol/l)	60.4(27.7)	78.5(30.2)	<0.0001	81.4(43.0)	116.8(54.3)	<0.0001
fasting plasma glucose (mmol/L)	5.5(1.2)	6.4(2.3)	0.0012	5.0(0.7)	5.6(1.4)	<0.0001
fasting serum insulin (mU/l)	7.9(5.3)	20.8(11.8)	<0.0001	6.2(3.3)	14.8(8.0)	<0.0001
HOMA-IR	2.0(2.0)	6.1(4.3)	<0.0001	1.4(0.9)	3.8(2.9)	<0.0001
Cell volumn (pl)	503(142	826(192)	<0.0001	474(179)	862(180)	<0.0001
Spontaneous lipolys (mmol glycerol/2hrs/ESAT)	0.8(0.7)	3.8(2.7)	<0.0001	1.0(0.9)	3.4(2.9)	<0.0001
Adipose morphology (pl)	“−6(128)”	13(171	0.36	“−13(149)	10(168)	0.054

Where: obese is defined as BMI>30 kg/m^2^; Spontaneous lipolysis rate was calculated as glycerol release divided by the lipid weight of the incubated fat cells; adipose morphology is difference between the measured and expected mean adipocyte volume at the abdominal subcutaneous adipose tissue depot. continuous variables are presented as mean (SD), groups were compared with Student’s *t*-test (unpaired). BMI, body mass index, WHR, waist hip ratio, HDL, high-density lipoprotein, HOMA-IR, homeostatic model assessment of insulin resistance, ESAT, estimated abdominal subcutaneous fat mass.

**Table 2 cells-09-01085-t002:** Tag SNPs associated with adipose morphology with *p* < 10^−5^.

Chrom	POS	ID	Effect Allele	BETA	L95	U95	*P*	A1_FREQ
1	3297459	rs201839757	I	198	113	283	5.75E-06	0.0126
1	48004858	rs12032932	T	167	96	238	4.40E-06	0.0110
1	76545413	rs10443175	T	−40	−57	−22	8.15E-06	0.2247
1	157782519	rs77346326	T	136	81	191	1.28E-06	0.0203
2	19739010	rs3914966	T	−34	−48	−19	4.64E-06	0.4256
2*	38508250	rs147711728	T	164	93	234	6.30E-06	0.0108
2	170990253	rs185528286	C	159	91	226	4.45E-06	0.0153
3*	189410000	rs2378515	G	54	34	73	5.29E-08	0.1802
4	170028965	rs202156267	A	−164	−229	−99	9.12E-07	0.0313
6	97632917	rs148707864	G	140	79	200	7.01E-06	0.0148
6*	130581980	rs62431222	A	102	61	144	1.36E-06	0.0329
7	52698748	rs150804725	I	−112	−159	−64	4.65E-06	0.0297
7	156412814	rs849073	G	−34	−48	−19	9.16E-06	0.4351
8*	1784364	rs11988258	A	−34	−48	−20	4.06E-06	0.4201
8	32613177	rs75468268	G	155	87	223	8.95E-06	0.0134
8	69224040	rs814465	C	82	49	114	1.04E-06	0.0537
8	141147356	rs142956251	A	155	87	224	8.43E-06	0.0139
9*	21340235	rs145072648	C	163	97	230	1.59E-06	0.0126
9	30859764	rs138591003	A	−178	−255	−100	7.76E-06	0.0108
10	97277536	rs189712177	G	186	107	265	4.43E-06	0.0115
10	128606691	rs71490795	A	121	69	174	6.52E-06	0.0225
11	23509085	rs148697259	A	133	75	192	9.22E-06	0.0181
11	99369450	rs72991567	G	67	39	95	2.54E-06	0.0804
12	73510344	rs60148685	D	34	19	48	8.09E-06	0.4437
13	30571045	rs200421910	I	−64	−92	−36	8.33E-06	0.0988
13	109383269	rs151235076	T	188	108	267	4.27E-06	0.0097
16	2390565	rs45501400	T	169	94	243	9.89E-06	0.0163
17*	22020568	rs201766885	C	113	65	161	4.24E-06	0.0464
19	53280252	rs10414169	T	131	74	189	8.82E-06	0.0183
22	32558640	rs144964180	A	160	90	229	7.28E-06	0.0122
22	49801423	rs9627723	C	45	26	64	4.15E-06	0.3654

* Tag-SNP in regions with ≥ 3 SNPs with *p* < 10^−5^ for association with adipose morphology. Only results for SNPs with minor allele frequency (MAF)>1% are shown. I = insertion; D = deletion. SNP, single nucleotide polymorphism, Chrom, chromosome, POS, position in the genome, ID, reference SNP accession number, BETA, beta coefficient, L95, lower 95% confidence interval, U95, upper 95% confidence interval, A1_FREQ, frequency of effect allele.

**Table 3 cells-09-01085-t003:** Tag genome-wide association study (GWAS) SNPs associated with type 2 diabetes and with adipose morphology in the current study.

			GWAS Adipose Morphology				GWAS Catalogue T2D				Consistent Risk Alleles For
SNP	CHROM	POS	A1	A1_FREQ	BETA	*P*	PUBMED	Reported Genes	Risk Allele	OR BETA	Adipose Hypertrophy and T2D
rs12031188	1	51103268	C	0.40	15	4.50E-02	30718926	*FAF1*	C	1.08	yes
rs58432198	1	51256091	T	0.11	−24	4.58E-02	30718926	*FAF1*	C	1.10	no
rs2088315	3	162382517	G	0.50	16	3.27E-02	30130595	*OTOL1, LINC01192*	?	5.26	not reported
rs8192675	3	170724883	C	0.30	17	4.13E-02	28566273	*NR*	T	1.05	no
rs138306797	3	185545719	T	0.02	76	1.32E-02	25760438	*IGF2BP2*	?	0.78	not reported
rs9379084	6	7231843	A	0.10	−48	2.94E-04	29632382	*RREB1*	G	1.09	yes
rs9460550	6	20719561	A	0.16	26	1.01E-02	31118516	*CDKAL1*	A	1.15	yes
rs72892910	6	50816887	T	0.20	−18	4.61E-02	30054458	*TFAP2B*	T	0.06	yes
rs9384193	6	154554249	C	0.38	−16	3.97E-02	28060188	*OPRM1, CNKSR3*	?		not reported
rs10231619	7	43320594	T	0.16	41	7.18E-05	25102180	*HECW1*	T	1.13	yes
rs4729854	7	102383663	A	0.47	−18	2.00E-02	30595370		?		not reported
rs896854	8	95960511	C	0.47	20	8.30E-03	20581827	*TP53INP1*	T	1.06	no
											no
rs10761745	10	65101071	G	0.14	−28	9.68E-03	28406950	*JMJD1C*	?	1.20	not reported
rs4929965	11	2197286	A	0.37	16	4.17E-02	30718926	*INS, IGF2*	A	1.12	yes
rs2237892	11	2839751	T	0.07	−47	1.12E-03	22961080	*KCNQ1*	C	1.32	yes
rs2722769	11	11228374	G	0.42	−16	3.48E-02	22238593	*GALNTL4, LOC729013*	C	1.35	unclear
rs7931302	11	128236058	C	0.29	19	1.92E-02	30054458	*ETS1*	C	0.05	yes
rs11048456	12	26463082	C	0.26	−17	4.66E-02	30054458	*ITPR2*	C	0.05	yes
rs1153188	12	55098996	T	0.26	21	1.79E-02	18372903	*DCD*	A	1.08	no
rs730570	14	101142890	G	0.18	−20	3.74E-02	21573907	*C14orf70*	G	1.14	no
rs1436955	15	62404382	T	0.28	−17	3.62E-02	20862305	*C2CD4B*	C	1.13	yes
rs781852	17	3953102	G	0.36	17	3.43E-02	29632382	*ZZEF1*	G	1.05	yes
rs11873305	18	58049192	C	0.04	54	1.04E-02	22885922	*MC4R*	A	1.18	no
rs7274168	20	32435978	C	0.41	−20	8.28E-03	30595370		?		not reported
rs6017317	20	42946966	G	0.23	17	4.06E-02	22158537	*FITM2, R3HDML, HNF4A*	G	1.09	yes
rs16988991	20	42989777	A	0.20	17	4.63E-02	30718926	*HNF4A*	A	1.05	yes
rs2833610	21	33385186	A	o	-17	2.75E-02	21490949	*HUNK*	A	1.17	no

Where A1 = effect allele; GWAS, genome-wide association study, T2D, type 2 diabetes, SNP, single nucleotide polymorphism, CHROM, chromosome, POS, position in the genome, A1_FREQ, frequency of effect allele, BETA, beta coefficient, OR, Odds Ratio.

## Data Availability

Summary GWAS is available upon request.

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
