# Peer review of "Genome-Wide Association Study of Diabetogenic Adipose Morphology in the GENetics of Adipocyte Lipolysis (GENiAL) Cohort"

_cells, 2020, doi:10.3390/cells9051085_

Round 1

Reviewer 1 Report

Overall, the manuscript entitled “Genome-wide association study of adipocyte lipolysis in the GENetics of adipocyte lipolysis (GENiAL) cohort” provides novel application of GWAS methodology for analyzing specific markers expression in adipocytes from Type 2 diabetes patients. The topic is current and very interesting. Minor issues need revision.

Specific comments:
- Abstract: Good concision and clarity. In line 36, please add a dot after brackets.
- Introduction: This chapter has enough information about the subject of the study.
- Methodology: Enough description of applied methods are included in the text. Please, explain the meaning of abbreviation MAF (line 105).
- Results: In line 194, please change expression "non-protein coding regions" for other more comprenhensible.
- Discussion: This chapter is well written and comparison with other published works is well balanced.

- Tables: In table 1, separation of column name to "non obese" is suggested.
- Figures:
+ In figure 2A, a legend to explain colors in bars would be desireable.
+ In figure 4C, please correct letter C at the top left of the plot.

Reviewer 2 Report

Hypertrophic adipose morphology has been associated with development of insulin resistance and diabetes. The aim of this work was to identify new candidate genes impacting adipocyte development using a genome-wide association study in the GENiAL cohort comprising 948 participants for which subcutaneous adipose tissue had been previously measured. They reported that 66 SNPs in 31 loci showed suggestive association with adipose morphology. Their subsequent analysis was focused on six genetic loci <1 Mb and containing at least 3 SNPs. Transcriptome profiling in the six genetic loci revealed six genes which were expressed in adipocytes. Functional evaluation of these genes in adipose derived precursor cells indicated their involvement in adipocyte differentiation.

This paper is interesting. However, I have some criticisms.

Criticisms:

1- Fig. 3B the authors showed a decrease of bodipy quantification in hMSCs cells. However, since these are relative values compared to the control, we have no idea of the true level of lipid accumulation in these cells. Can the authors show the cell staining?

2- Fig. 3C the authors showed a decrease of glycerol amount in the medium. However, since these are relative values compared to the control, we have no idea of the true level in these cells. Can the authors show the amount measured?

3- You have more cells (Fig 4) and less lipids (Fig 3) in your treated cells regardless of the siRNAs used. Is this because there is less differentiation or is it because they induce smaller adipocytes? The photos at 9 days of differentiation may shed some light on this important point.

4- In your cohort of 114 women you found an inverse relationship between ATL2 expression and adipose morphology, and a positive association between CYP1B1 and adipose morphology, whereas no association were observed for ARHGEF10, TMEM200A, L3MBTL3 and C17orf51. In this basis, you assumed that CYP1B1 and/or ATL2 as potential causal links between genetic variants and adipose morphology.

However, your functional studies using siRNAs do not suggest differences between the 6 genes analyzed. How do you explain this on the basis of these initial mechanistic studies?

5- In the discussion you develop a large paragraph about the TP63 gene when it is not expressed in your adipocytes. What's the point?

Round 2

Reviewer 2 Report

  What do you want to do ? New mailCopy The authors responded satisfactorily to the questions asked.   What do you want to do ? New mailCopy     What do you want to do ? New mailCopy